# Comparison of de-duplication methods used by WHO Global Antimicrobial Resistance Surveillance System (GLASS) and Japan Nosocomial Infections Surveillance (JANIS) in the surveillance of antimicrobial resistance

Toshiki Kajihara[1][☯]*, Koji Yahara[1][☯]*, John Stelling[2], Sergey Romualdovich Eremin[3], Barbara Tornimbene[3], Visanu Thamlikitkul[4], Aki Hirabayashi[1], Eiko Anzai[1], Satoyo Wakai[1], Nobuaki Matsunaga[5], Kayoko Hayakawa[6], Norio Ohmagari[5], Motoyuki Sugai[1], Keigo Shibayama[1,7]

1 Antimicrobial Resistance Research Center, National Institute of Infectious Diseases, Higashimurayama, Tokyo, Japan, 2 Brigham and Women's Hospital and Harvard Medical School, Boston, Massachusetts, United States of America, 3 WHO Headquarters, Geneva, Switzerland, 4 Division of Infectious Diseases and Tropical Medicine, Department of Medicine, Faculty of Medicine Siriraj Hospital, Mahidol University, Bangkok, Thailand, 5 AMR Clinical Reference Center, National Center for Global Health and Medicine Hospital, Toyama, Tokyo, Japan, 6 Disease Control and Prevention Center, National Center for Global Health and Medicine Hospital, Toyama, Tokyo, Japan, 7 Department of Bacteriology II, National Institute of Infectious Diseases, Musashimurayama, Tokyo, Japan

☯ These authors contributed equally to this work.
* kajihara@nih.go.jp (TK); k-yahara@nih.go.jp (KY)

## Abstract

A major issue in the surveillance of antimicrobial resistance (AMR) is "de-duplication" or removal of repeated isolates, for which there exist multiple methods. The World Health Organization (WHO) Global Antimicrobial Resistance Surveillance System (GLASS) requires de-duplication by selecting only the first isolate of a given bacterial species per patient per surveillance period per specimen type per age group, gender, and infection origin stratification. However, no study on the comparative application of this method has been reported. The objective of this study was to evaluate differences in data tabulation between the WHO GLASS and the Japan Nosocomial Infections Surveillance (JANIS) system, which counts both patients and isolates after removing repeated isolates of the same bacterial species isolated from a patient within 30 days, regardless of specimen type, but distinguishing isolates with change of antimicrobial resistance phenotype. All bacterial data, consisting of approximately 8 million samples from 1795 Japanese hospitals in 2017 were exported from the JANIS database, and were tabulated using either the de-duplication algorithm of GLASS, or JANIS. We compared the tabulated results of the total number of patients whose blood and urine cultures were taken and of the percentage of resistant isolates of *Escherichia coli* for each priority antibiotic. The number of patients per specimen type tabulated by the JANIS method was always smaller than that of GLASS. There was a small (< 3%) difference in the percentage of resistance of *E. coli* for any antibiotic between the two methods in both out- and inpatient settings and blood and urine isolates. The two tabulation methods

**Data Availability Statement:** All data to replicate the study's findings are fully available at https://www.niid.go.jp/niid/ja/from-lab/2415-amrc/9213-janis-glass-excel-en.html as written in the main text.

**Funding:** This study was supported by Research Program on Emerging and Re-emerging Infectious Diseases from the Japan Agency for Medical Research and Development (AMED) under grant number JP19fk010806 and a Health and Labour Sciences Research Grant from the Ministry of Health, Labour, and Welfare, Japan (H30-shokuhin-ippan-006). This study was also supported by the Research Program on Emerging and Re-emerging Infectious Diseases from the Japanese Agency for Medical Research and Development (AMED) under grant number 19fk0108061j0002 to KS and the National Institutes of Health (NIH) grant number U01CA207167 to JS. Its contents are solely the responsibility of the authors and do not necessarily represent the official views of AMED or NIH.

**Competing interests:** The authors have declared that no competing interests exist.

did not show considerable differences in terms of the tabulated percentages of resistance for *E. coli*. We further discuss how the use of GLASS tabulations to create a public software and website that could help to facilitate the understanding of and treatment against AMR.

## Introduction

Antimicrobial resistance (AMR) is a global public health threat and its surveillance is of vital importance as underscored by the development of the Global Action Plan on AMR launched by the World Health Organization (WHO) in 2015 [1]. AMR could result in 10 million deaths per year by 2050 [2, 3] and incur high economic costs [3, 4]. The WHO Global Action Plan 2015 emphasizes the need to develop surveillance plans to track AMR across the globe. To enable this, proper information technology (IT) infrastructure is essential at both the national and international level [5, 6]. In an effort to facilitate this, the WHO created the Global Antimicrobial Resistance Surveillance System (GLASS) in October 2015 that aims to enable international comparison, analysis, and sharing of AMR data by encouraging countries to submit their national AMR surveillance data via the internet in a unified manner [7, 8].

One of the largest IT-based national AMR surveillance systems in the world is the Japan Nosocomial Infections Surveillance (JANIS) program, which centralizes AMR related hospital data and allows it to be compiled, analyzed, and published as not only national AMR surveillance reports but also benchmarking feedback reports for each participating hospital to facilitate infection control practices [6, 9]. JANIS became completely web-based in 2007, and the number of hospitals participating in the Clinical Laboratory module has steadily increased to 2120 hospitals (of the approximately 8,000 hospitals across Japan) at the beginning of 2019. Notably, the Clinical Laboratory module collects all routine bacteriological test results. JANIS started approximately 10 years earlier than GLASS and has its own rules and methods for the tabulation of data. National AMR data from Japan in 2017 were published in GLASS Early implementation reports [8] based on data used to generate the annual open report for JANIS (JANIS English website https://janis.mhlw.go.jp/english) using its original tabulation method. To allow for international comparison, it is necessary to clarify the differences between the tabulation methods used by various AMR surveillance databases in an effort to reduce their impact on national AMR statistics.

A major difference between these systems lies in their "de-duplication" method that removes repeated isolates of the same bacterial species from the same patient over time so that duplicate findings are excluded before generating the aggregated data. There are a number of approaches for "de-duplication", and there is no single "ideal" approach; however, some have been shown to have greater feasibility than others. These data can be used for different purposes such as infection prevention and control purposes, and generating an annual AMR surveillance report used for treatment guidelines or facility benchmarking. In previous studies, Kohlmann *et al* (2016) reported the differences between various de-duplication methods. Their report indicated differences in resistance rates of up to five percent for *Staphylococcus aureus*, *Escherichia coli*, and *Klebsiella pneumoniae* and up to ten percent for *Pseudomonas aeruginosa* between the three different methods: 1) selecting all isolates (i.e., without de-duplication), 2) different episode-based de-duplications (minimal time interval: 2, 5, 10, 30, or 100 days) and 3) de-duplication selecting the first isolate for each patient [10]. The European Society of Clinical Microbiology and Infectious Diseases (ESCMID) Study Group for

Antimicrobial Resistance Surveillance (ESGARS) also reported similar results when they did their comparisons [11].

The Clinical & Laboratory Standards Institute (CLSI) also mentions that de-duplication is defined as "the inclusion of only the first isolate of a given species per patient per analysis period," for any full dataset of data subject, *i.e.* the third de-duplication method mentioned above [10]. WHO GLASS chose the "first isolate" or "one isolate per patient" approach to create the internationally standardized reporting structure, consistent with CLSI recommendations, for several reasons, including feasibility (not all the countries can apply even this simple algorithm) and the fact that many countries already have been using this approach when reporting to international networks, including the countries enrolled in the European Antimicrobial Resistance Surveillance Network (EARS-Net) [12] and the Central Asian and Eastern European Surveillance of Antimicrobial Resistance (CAESAR) network [13].

Consistent with the networks mentioned above, WHO GLASS recommends the simple "one isolate per patient" tabulation stratified by specimen type, and further data stratification for gender, age group, and infection origin [7]. GLASS also requests each country to report the number of patients tested including both positive and negative (no growth) samples classified according to gender, age group and infection origin (community or hospital) after de-duplication and specimen (BLOOD, URINE, STOOL, and GENITAL swab) stratification. However, no study has reported a comparison between the de-duplication method from the WHO GLASS system and any other method. Here, we conducted a comparison using the national AMR data from Japan (2017) stored in the JANIS database. Basically, JANIS uses a method that removes repeated isolates of the same species isolated from a patient within 30 days, regardless of specimen type, but still distinguishing isolates that showed change of antimicrobial resistance phenotype (e.g., from susceptible to resistant for any antibiotic, as detailed in Materials and methods). Therefore, the de-duplication method from JANIS is conceptually different from that of GLASS.

Our systematic and quantitative comparison revealed the extent of these differences and illustrated how these differences could influence the surveillance statistics including the total number of patients and percentage of resistance for each bacterial species with respect to different classes of antimicrobial drugs.

## Materials and methods

### Data preparation, tabulation, and analysis

Data from approximately 8 million samples collected during 2017 were exported from the JANIS database as a text file. This file was sorted according to facility ID, patient ID, specimen collection date, inpatient or outpatient, and specimen ID.

The file was tabulated using the de-duplication algorithm from either JANIS or WHO GLASS (see below for detail) using an in-house Java program to create two files required for WHO-GLASS submission [7]: 1) *Sample* file containing the number of patients from whom GLASS priority specimens were taken, stratified by specimen type, gender, infection origin (inpatient or outpatient), and age group; 2) *RIS* file with susceptibility testing results containing the number of resistant, intermediate and susceptible isolates (and other interpretations of antimicrobial susceptibility testing (AST) results), stratified by the same variables as in the *Sample* file. Among the "other interpretations of AST results", GLASS requires a calculation of the number of isolates with AST results not reported or not performed for each antibiotic (indicated as UNKNOWN_NO_AST variable in GLASS) in order to estimate the magnitude of selective testing. The in-house Java program also implemented a function to calculate this

value. Because age and gender are input fields in WHO-GLASS but not in JANIS, a subset of the data that did not have age or gender information was excluded from the tabulation.

The two types of tabulated files were further processed to calculate summations across genders and according to new age groups (<15, 15–64, 64<) rather than those defined by GLASS (<1, 1–4, 5–14, 15–24, 25–34, 35–44, 45–54, 55–64, 65–74, 75–84, 84<) using in-house Perl scripts.

The in-house Java program and Perl scripts are available at https://github.com/bioprojects/GLASS-JANIS-comparison.

In the graphs and tables, names of the antibiotics have been abbreviated using the WHO-GLASS guidelines and appear as follows: ampicillin = AMP, imipenem = IPM, meropenem = MEM, cefotaxime = CTX, ceftriaxone = CRO, ceftazidime = CAZ, cefepime = FEP, levofloxacin = LVX, ciprofloxacin = CIP.

**Comparison of de-duplications between WHO-GLASS and JANIS.** Both the *Sample* and *RIS* files were generated from the same database for the specific purpose of reporting to GLASS. De-duplication is performed on the source database as follows. For the counting of the number of patients in the *Sample* file, GLASS requires selection and reporting of only the first result from bacterial culture during each surveillance period (e.g. 12 months) for each patient in each specimen type per age group, gender, and infection origin stratification [7]. On the other hand, JANIS' server-side program selects the first result from each patient during a 30 day cycle with each resistance phenotype obtained by AST irrespective of specimen type, given that a subsequent bacterial culturing after 30 days usually corresponds to a different episode given the average length of a hospital stay (< 30 days) in Japan. Thus, JANIS targets infected/non-infected patients, independent from the anatomical site of infection.

To count the number of resistant, intermediate and susceptible isolates (and other interpretations of AST results) for each bacterial species, as in the *RIS* file, GLASS requires the same de-duplication as before: selection and reporting of only the first result of AST for each surveillance period for each patient per specimen type and data stratification and bacterial species [7]. To count the isolates used in antibiograms constructed for each surveyed species in the national and feedback reports from JANIS, its server-side program also conducts the same de-duplication using the 30 day rule, but distinguishes isolates with change of antimicrobial resistance phenotype obtained by AST as different isolates even if they were isolated within 30 days from the same patient [9, 14]. A subsequent isolate is selected and counted if it shows phenotypic change from susceptible to resistant (or vice versa) or a 4-fold or more (or 4-fold less) comparison of MIC value for a specific antibiotic when compared to a previous isolate from the same patient within the 30-day period, as implemented in the in-house Java program. The server-side program of JANIS also selects the 1st isolate with intermediate susceptibility to a specific antibiotic isolated from the same patient within the 30-days period, although the intermediate susceptibility is not accounted for in this study because it focuses on and compares percentage of resistance.

These differences are summarized in Table 1.

**Table 1. Differences in the de-duplication methods from GLASS and JANIS.**

|  | GLASS | JANIS (to count patients) | JANIS (to count isolates in the antibiograms) |
|---|---|---|---|
| Time period in which the de-duplication is conducted | surveillance period (e.g. 12 months) | 30 days | 30 days |
| Accounting for specimen types | Yes | No | No |
| Accounting for difference in AST results | No | No | Yes |

### Ethical considerations

Patient identifiers were de-identified by each hospital before data submission to JANIS. The anonymous data stored in the JANIS database were exported and analyzed following approval by the Ministry of Health, Labor and Welfare (approval number 1115–3) according to Article 32 of the Statistics Act.

## Results

National AMR surveillance data of all bacterial culturing and AST results collected at 1795 Japanese hospitals in 2017 were tabulated according to the procedure defined by WHO-GLASS or JANIS, respectively. The number of tabulated patients whose blood and urine cultures were taken is shown stratified by inpatient, outpatient, and age group in Fig 1A and 1B, respectively. In the former, both the total number of outpatients and inpatients de-duplicated by the GLASS method (red bars in the "Total" in Fig 1A) were approximately 140,000 and 200,000 patients more than those included using the JANIS method (blue bars in "Total" in Fig 1A). GLASS counts the number of patients infected per anatomical site (i.e. the number of cases of different types of infection, for example bloodstream and urinary tract infection). The discrepancy between JANIS and GLASS probably results from the fact that JANIS only considers the site of the first occurrence of the pathogen, but ignores the others (presuming that the following isolates have approximately the same resistance characteristics, so we can ignore them). Similarly, in the latter, the total number of outpatients and inpatients de-duplicated using the GLASS method (red bars in "Total" in Fig 1B right) were approximately 140,000 and 280,000 more than those included when the data were interpreted using the JANIS method (blue bars in "Total" in Fig 1B right). For both blood and urine, the difference was at least 140,000 larger in the inpatients than in outpatients. When the difference was examined between the three age groups (<15, 15–64, 64<), it was largest in the ≥65 years old group irrespective of outpatient, inpatient, and specimen type: the number of patients de-duplicated using the GLASS method was approximately 90,000–200,000 more than that of JANIS (blue vs red bars in "64<" in Fig 1A and 1B). An explanation for this may be that older patients tend to have multiple specimen types such as blood, urine, and sputum.

The number of patients per specimen type tabulated by the JANIS method was always smaller than those created using the GLASS method as the former conducts de-duplication of isolates of the same species isolated from a patient within 30 days, *regardless of specimen type*: when an isolate is found in urine and another was found in the blood (or vice versa) within 30 days, JANIS selects only the 1st isolate and ignores the subsequent isolate, which results in decreased numbers of patients per specimen type.

Next, comparison of the percentages of resistant isolates (using the total number of isolates with AST results for each antibiotic separately as the denominator) stratified by outpatient, inpatient, and specimen type in *E. coli* (as a representative species that is estimated to cause the largest number of cases and deaths due to AMR in Europe [15]) tabulated using the GLASS or JANIS methods are shown in Fig 2. JANIS method can use either just the 30 day rule to count patients ("JANIS (patients)" in blue in Fig 2), or it can account for the AST results and count isolates by distinguishing those with change of antimicrobial resistance phenotype obtained by AST ("JANIS (isolates)" in light blue in Fig 2, see Materials and methods for the detail). In respect of the *E. coli* blood isolates, there was little difference in resistance to any of the surveyed antibiotic when using the GLASS method between either the outpatient or inpatient settings (Fig 2A and 2B). The largest difference among the three groups was seen in the resistance to levofloxacin (LVX), with almost 1.8% resistance in outpatients and 1.6% in inpatients. For

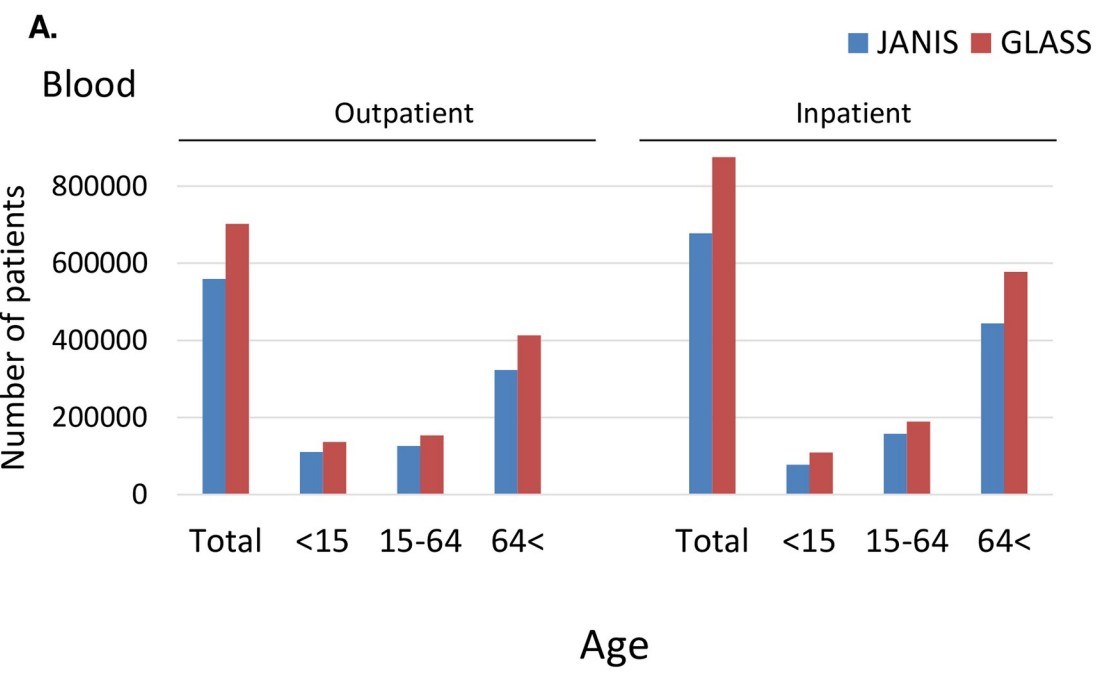

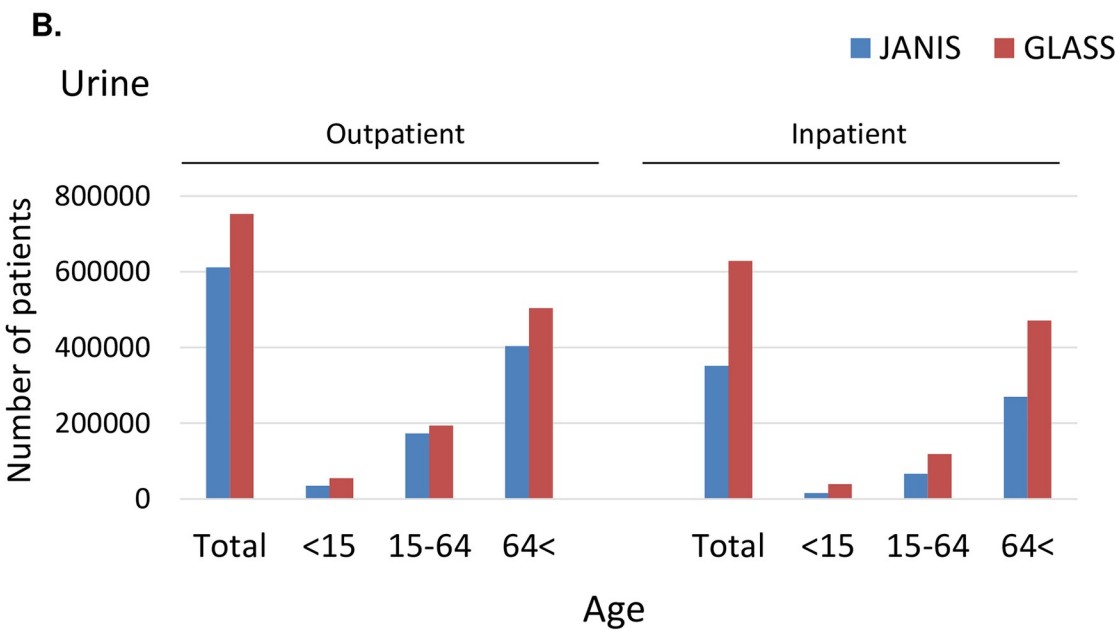

**Fig 1. Comparison of the number of patients with samples (blood and urine) collected for bacteriological testing.** The value was determined from the tabulated numbers for NUMSAMPLEDPATIENTS in the *Sample* file required for submission to GLASS and is shown separately for outpatient (left) and inpatient (right). The x-axis denotes age group and "Total" indicates the sum of all three age groups. The blue and dark red bars are the outcomes from JANIS and GLASS, respectively. (A) Blood. (B) Urine.

the *E. coli* urine isolates, the largest difference in resistance was similarly seen in levofloxacin (LVX), with almost 2.7% in outpatients and 2.8% in inpatients (Fig 2C and 2D). Comparison between the method for counting patients and that for counting isolates from JANIS showed that the former ("JANIS (patients)" colored in blue in Fig 2) kept the resistance values more

## E. coli from blood

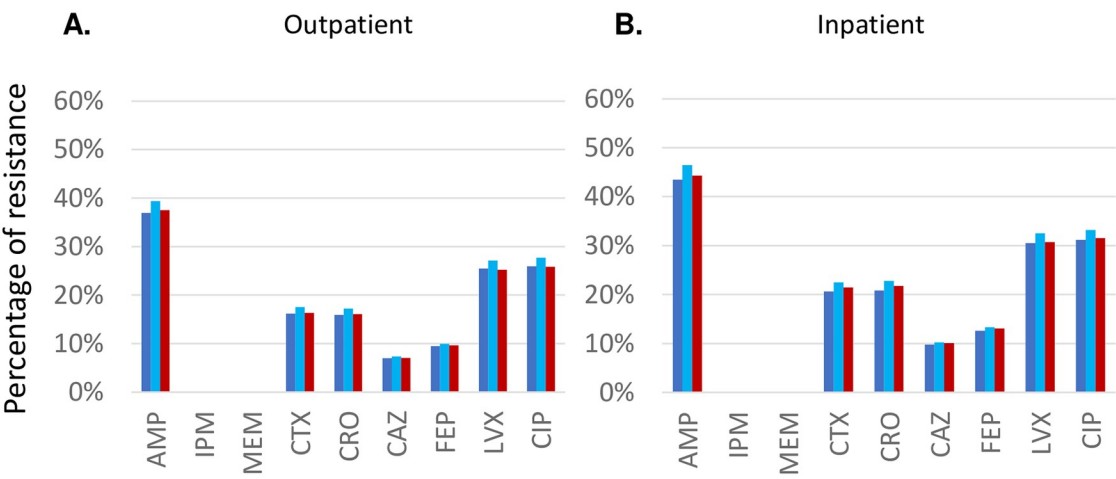

## E. coli from urine

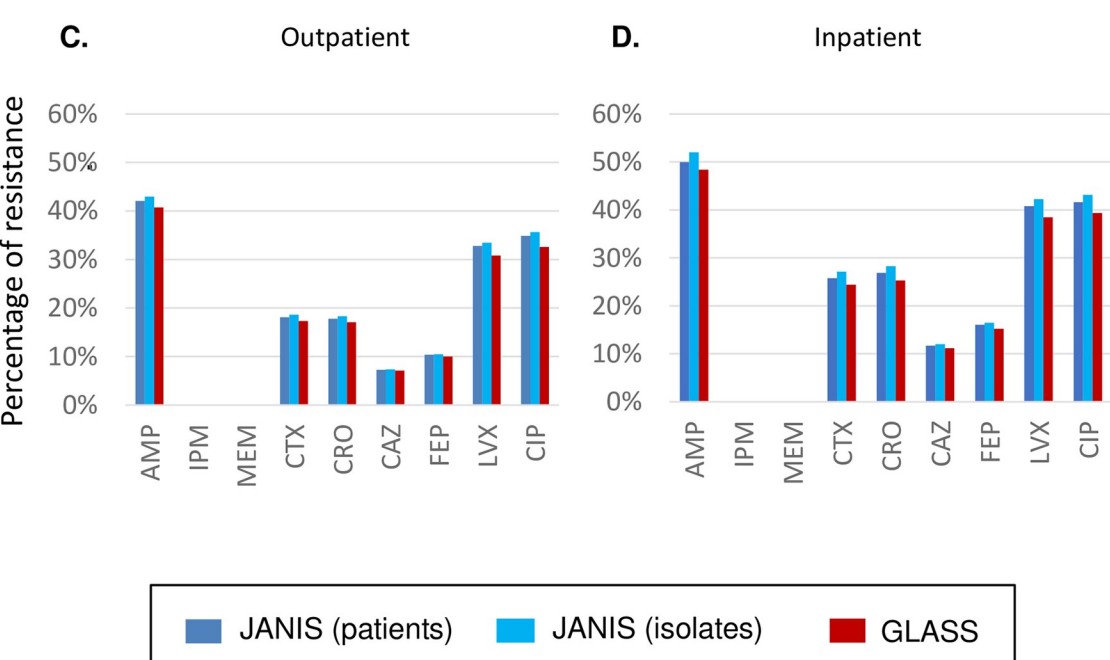

**Fig 2. Comparison of percentages of resistant *E. coli*.** Each value was calculated as the number of resistant isolates divided by the total number of isolates tested for an antimicrobial drug in the RIS file required for submission to GLASS, and is shown separately for outpatient (left) and inpatient (right). The blue and light blue bars indicate the number of patients and isolates tabulated using the de-duplication method from JANIS (see Materials and methods for detail). The dark red bars indicate the number of patients (that have 1-to-1 correspondence to the isolates) using the de-duplication method from GLASS. **(A-B) Blood.** The denominator for each antibiotic is always > 80,000 in outpatient and > 60,000 in inpatient settings, respectively. **(C-D) Urine.** The denominator for each antibiotic is always >10,000 in both outpatient and inpatient settings.

similar to the GLASS outcomes than the latter ("JANIS (isolates)" colored in light blue in Fig 2). This is because the latter conducts an additional selection and counting step to distinguish isolates with change of antimicrobial resistance phenotype obtained by AST, but both the JANIS method for counting the patient and that of GLASS do not. The difference is, however, small and at most 2.8%. Even if we stratified the data using the three age groups as in Fig 1, we found the same result and the difference was at most 3.2%.

In a voluntary surveillance like JANIS, the total number of isolates subject to AST is not equal between participating hospitals because of the variability in manners and procedures for routine AST between institutions. In this situation, GLASS requires a calculation of the number of isolates without AST results (indicated as UNKNOWN_NO_AST variable, see also Materials and methods). The percentages shown in Fig 2 were calculated without including this parameter, but we examined to what extent resistance changed following its inclusion (yellow in Fig 3 compared to red). The inclusion of these data increases the denominator and of course decreases the rate of resistance, but the extent was almost always less than 5%. In the case of ciprofloxacin (CIP), however, the inclusion of these data decreased the resistance rates for blood isolates from 16.4% to 13.6% and 20.6% to 17.4% in urine isolates in outpatient and inpatient settings, respectively. This is because many AST results were not reported or not performed for CIP, which accounts for > 30% of the total denominator. To account for these points, the GLASS report shows percentages of resistance without including UNKNOWN_NO_AST in the denominator (as in Fig 2 from this study) but marks an antibiotic like CIP as "> 30% unknown AST results" using a different color.

## Discussion

Patients with infectious diseases are usually subject to multiple rounds of specimen sampling for diagnostic or follow up evaluations, particularly in the case of antimicrobial resistant bacteria. Inclusion of all samples would lead to an overestimation of the proportion of resistance in a given population. De-duplication methods aim to reduce such bias. Our comparison of the de-duplication methods used for national surveillance between WHO GLASS and JANIS revealed little differences in the tabulated percentages of resistance to any antibiotic surveyed by GLASS, although WHO GLASS takes the difference in specimen type into account but JANIS does not.

Meanwhile, the comparison showed considerable differences in the number of tabulated patients per specimen type. The difference in the total number of patients per specimen type tabulated by the two methods was larger in the inpatient samples probably because inpatients are usually subject to broader sampling than outpatients are. The de-duplication method from JANIS selects the first result every 30 days for each patient *irrespective of* specimen type, while GLASS selects the first AST result for each specimen type. When the difference was examined between three age groups (<15, 15–64, 64<), the difference was largest in the over 65 years old group probably because it is more common to collect different types and number of specimens from patients aged over 65 years old.

The tabulated number of patients per specimen type is used in the GLASS report as a denominator for calculating frequency of infection of specific AMR bacteria in the "Non-susceptible pathogen-antimicrobial combination frequency" and "Non-susceptible pathogen-meropenem combination stratified frequency" sections [8]. The de-duplication method used by JANIS makes the number of patients for the specimen type lower, corresponding to higher frequencies of infection (per 100,000 tested patients) if the numerator is the same. However, the de-duplication method used by JANIS will also make the numerator lower, because the pathogen for that specimen type will be excluded if it isn't the first pathogen. Therefore, the

## *E. coli* from blood

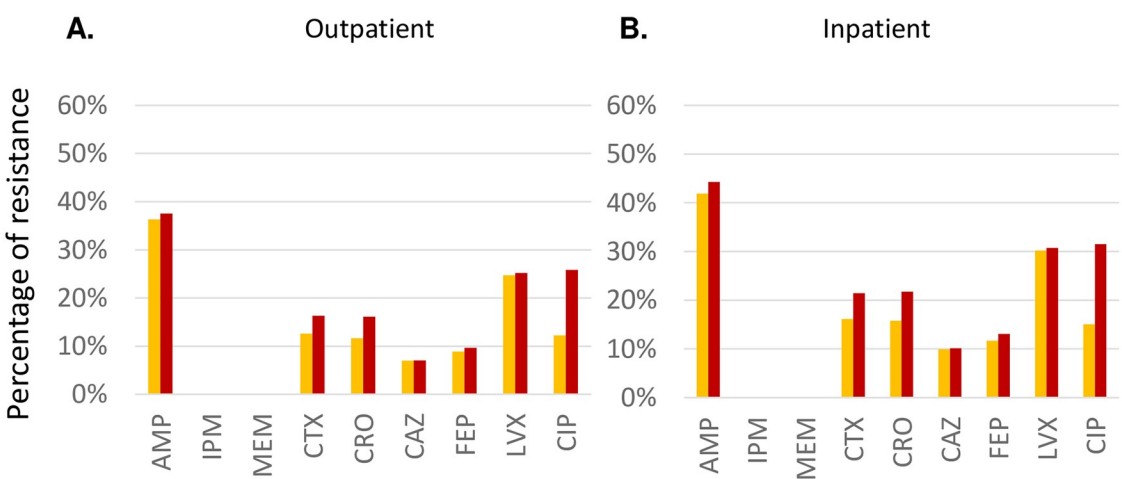

## *E. coli* from urine

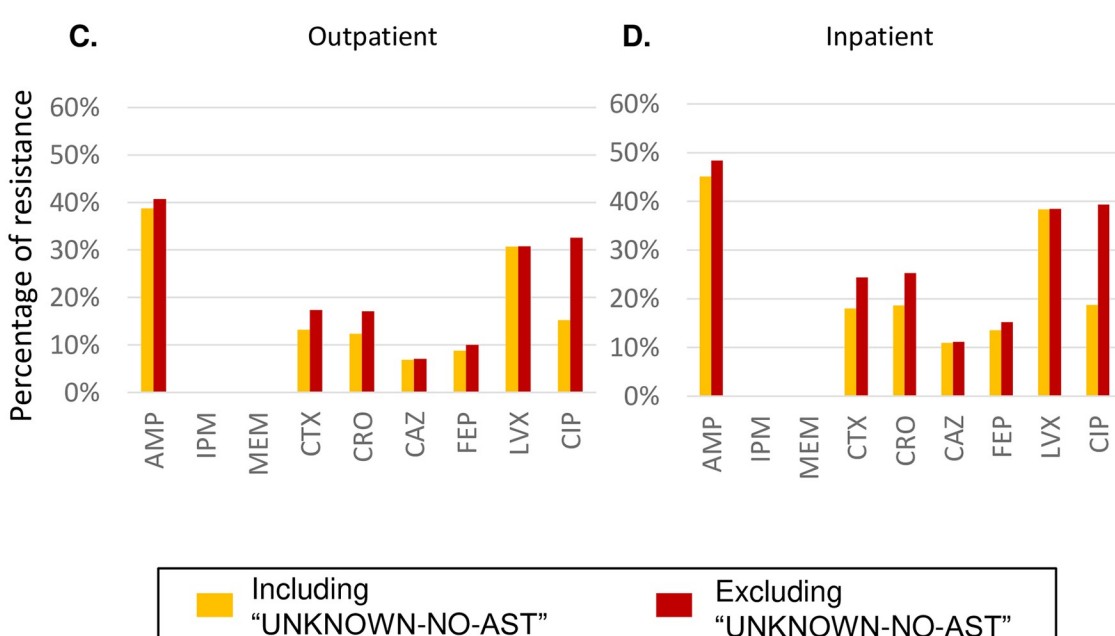

**Fig 3. Comparison of the percentage of resistant *E. coli* in various samples including or excluding UNKNOWN-NO-AST.** The number was calculated as the value for RESISTANT divided by the total number of isolates tested with or without "UNKNOWN-NO-AST" representing the number of isolates with AST results not performed or reported (colored in yellow or red) for an antimicrobial drug in the RIS file required for submission to GLASS, and is shown separately for outpatient (left) and inpatients (right). (A-B) Blood. (C-D) Urine.

frequency of specimen-type specific infection or colonization rate might or might not become higher.

National AMR data from Japan in 2017 were first included in the GLASS report (early implementation 2017–2018). These data were based on the dataset used for the annual open report from JANIS in 2017 and the tabulation method used was the one from JANIS. This study confirmed that there was no problem in terms of tabulated percentage of resistance, although it showed considerable differences in the number of tabulated patients per specimen type. Based on these results, Japan will prepare and submit data in 2018 after using the de-duplication method defined by GLASS.

The outpatients and inpatients in this study were tabulated from data related to strains isolated from patients before or after hospitalization, and do not always correspond to community-acquired or hospital-acquired infection, which may be more important in a clinical setting, but are very difficult to classify. For example, in the Clinical Practice Guidelines for Hospital acquired and Ventilator associated Pneumonia by Infectious Diseases Society of America and American Thoracic Society, hospital acquired pneumonia is defined as that development of infection 48 h after admission [16]. However, the admission date is not a mandatory field in JANIS, making it impossible to automatically classify community-acquired and hospital-acquired infections.

In a previous study, Kohlmann *et al*. compared the percentages of resistance tabulated from the first isolate, isolates selected after de-duplication assuming an episode of 100/ 30/ 10/ 5 or 2 days and all isolates [10]. Their study reported that there was small (at most a few percentage) difference in tabulated percentages of resistance between the first isolate rule and those calculated assuming a 30 day episode [10]. In another study [17], Hindler *et al*. reported the same kind of findings: 1) de-duplication is important, otherwise resistance rates become significantly higher; 2) the manner in which repeated isolates are removed is less important–they will lead to different estimates (predictably higher or lower depending on the method), but the differences are relatively small. They are consistent with our results that found almost no difference between the de-duplication methods from GLASS and JANIS. Further studies are warranted to examine whether the results are also applicable to other low- and middle-income countries with higher prevalence of AMR by collecting and analyzing data regarding deduplication of the isolates from AMR surveillance in such countries.

For data submission to WHO GLASS, each country must prepare the *RIS* file containing 15 variables (COUNTRY, YEAR, SPECIMEN, PATHOGEN, GENDER, ORIGIN, AGE GROUP, ANTIBIOTIC, RESISTANT, INTERMEDIATE, NON-SUSCEPTIBLE, SUSCEPTIBLE, UNKNOWN_NO_AST, UNKNOWN_NO_BREAKPOINTS, BATCH ID) and a *Sample* file containing 8 variables (COUNTRY, YEAR, SPECIMEN, GENDER, ORIGIN, AGE GROUP, NUMSAMPLEDPATIENTS, BATCH ID) [7]. WHO GLASS requests that submitters calculate the number of isolates with AST results not reported or not performed for a specific antibiotic (indicated as UNKNOWN_NO_AST), which is needed to estimate the magnitude of the bias related to selective testing. However, it is not included in the denominator used to calculate the percentage of resistance in the GLASS report. For example, in our study, the proportion of UNKNOWN_NO_AST was highest (approximately 50%) in ciprofloxacin among both blood and urine isolates. In the GLASS report, ciprofloxacin is indeed marked as "> 30% unknown AST results", but its percentage of resistance is almost the same as that of levofloxacin, after excluding the number of UNKNOWN_NO_AST isolates from the denominator. This suggests that the number of UNKNOWN_NO_AST should not greatly impact resistance estimates if there is no bias in which strains were or were not tested. On the other hand, if certain reserve agents are only tested as second-line agents, then this could introduce significant bias towards higher resistance estimates.

As a publicly available product of the present study, we have created an Excel tool that stores and can analyze the data tabulated using the GLASS de-duplication method (its graphical user interface is shown in S1 Fig). It can be used by simple mouse-clicking to generate stratified antibiograms according to GLASS, and is available at https://github.com/bioprojects/GLASS-JANIS-comparison and detailed in Supporting Information. Furthermore, we have tabulated the data from 2011 to 2017 for each prefecture using the de-duplication method from GLASS. The data were incorporated into the One health platform website (https://amr-onehealth-platform.ncgm.go.jp), in which each antibiotic can be individually evaluated for an annual trend regarding percentage of resistance, and its distribution among the prefectures and reported as a map.

In conclusion, our comparison of the de-duplication methods used in national AMR surveillance between the WHO GLASS and JANIS algorithms revealed a slight difference in the tabulated percentages of resistance, but considerable differences in the number of patients stratified by specimen type. Given these results, Japan will switch to the de-duplication method from GLASS stratified by each specimen type for the next submission of data in 2018 to GLASS, meanwhile annual reports of JANIS will continue to tabulate all specimen type together. This study significantly deepened our understanding of data tabulation in GLASS, which helped us to develop the publicly available Excel tool that can be used to generate stratified antibiograms and the website that may be broadly useful in understanding and combating AMR.

## Supporting information

**S1 Fig. Graphical user interface and usage of the excel tool.** It stores and can analyze the data tabulated using the de-duplication in GLASS. Users can interact with these data by simply clicking their mouse, and create antibiograms stratified by the bed size of hospitals (under 200, over 200 and under 500, over 500 beds), inpatient or outpatient, gender, specimen type, and age group as defined by GLASS. Based on this, users can create antibiograms after excluding UNKNOWN_NO_AST. As another variable regarding missing data, GLASS defines UNKNOWN_NO_BREAKPOINTS representing the number of isolates where AST was performed but no interpretation of the results is available for a specific antibiotic. In the Excel tool, we have renamed this as UNKNOWN_NO_SIR to imply that S-I-R interpretation was not possible. It can be included or excluded from the denominator for calculating resistance rates in the tool.
(DOCX)

## Acknowledgments

We are grateful to all the hospitals that participated and contributed data to JANIS, and to Editage (www.editage.jp) for English language editing.

## Author Contributions

**Conceptualization:** Toshiki Kajihara, Koji Yahara, Nobuaki Matsunaga.

**Data curation:** Toshiki Kajihara, Koji Yahara, Eiko Anzai, Satoyo Wakai.

**Formal analysis:** Toshiki Kajihara, Koji Yahara, Eiko Anzai, Satoyo Wakai.

**Funding acquisition:** John Stelling, Keigo Shibayama.

**Investigation:** Toshiki Kajihara, Koji Yahara, Eiko Anzai, Satoyo Wakai.

**Methodology:** Toshiki Kajihara, Koji Yahara.

**Project administration:** Toshiki Kajihara, Koji Yahara.

**Resources:** Toshiki Kajihara, Koji Yahara.

**Software:** Koji Yahara.

**Supervision:** Koji Yahara, John Stelling, Sergey Romualdovich Eremin, Barbara Tornimbene, Visanu Thamlikitkul, Motoyuki Sugai, Keigo Shibayama.

**Validation:** Toshiki Kajihara, Koji Yahara, Aki Hirabayashi.

**Visualization:** Toshiki Kajihara, Koji Yahara.

**Writing – original draft:** Toshiki Kajihara, Koji Yahara.

**Writing – review & editing:** Koji Yahara, John Stelling, Sergey Romualdovich Eremin, Barbara Tornimbene, Visanu Thamlikitkul, Aki Hirabayashi, Nobuaki Matsunaga, Kayoko Hayakawa, Norio Ohmagari, Motoyuki Sugai, Keigo Shibayama.

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
