## [Decision Letter · Decision Letter 0]

25 May 2020

PONE-D-20-00688

Comparison of de-duplication methods used by WHO Global Antimicrobial Resistance Surveillance System (GLASS) and Japan Nosocomial Infections Surveillance (JANIS) in the surveillance of antimicrobial resistance

PLOS ONE

Dear Dr. Kajihara,

Thank you for submitting your manuscript to PLOS ONE. After careful consideration, we feel that it has merit but does not fully meet PLOS ONE’s publication criteria as it currently stands. Therefore, we invite you to submit a revised version of the manuscript that addresses the points raised during the review process.

The reviewers were positive about the manuscript. There are some changes to be done to increase clarity. The manuscript should also be reduced in the introduction and discussion. That will increase its clearness and bring a clear message.

We look forward to receiving your revised manuscript.

Kind regards,

Patrick Butaye, DVM, PhD

Academic Editor

PLOS ONE

Journal Requirements:

2. Please amend the manuscript submission data (via Edit Submission) to include author Satoyo Wakai.

3. Please amend your authorship list in your manuscript file to include author Tomoyo Wakai.

Reviewers' comments:

Reviewer's Responses to Questions

**Comments to the Author**

1. Is the manuscript technically sound, and do the data support the conclusions?

Reviewer #1: Yes

Reviewer #2: Yes

2. Has the statistical analysis been performed appropriately and rigorously? 

Reviewer #1: Yes

Reviewer #2: Yes

3. Have the authors made all data underlying the findings in their manuscript fully available?

Reviewer #1: No

Reviewer #2: Yes

4. Is the manuscript presented in an intelligible fashion and written in standard English?

Reviewer #1: Yes

Reviewer #2: Yes

5. Review Comments to the Author

Reviewer #1: De-duplication of datasets used for surveillance of antimicrobial resistance is important and it has been shown that this strategy influences the final report. There are several strategies for de-duplication, the manuscript compares two of them, the Japanese JANIS and the WHO GLASS system. Depending on definition of the denominator for some antibiotics significant differences were found between the two methods. The methods and datasets appear comprehensive, presentation of results in general is adequate.

I only have few comments:

1. pg. 3 "per multi-resistance phenotype" . In the whole paper there is no definition of multi-resistence phenotype. It only multi-resistant strains are included, the result is severly skewed. Explain or define "multi-resistance".

2. pg. 8 line 2: "using"

3. pg 10 "considering antimicrobial resistance": in which form is resistance considered? This remains unclear in the whole manuscript.

4. pg 10 last line, pg 11 first line: usage of ~, -, <=, >= is not consistent in the whole manuscript, please change

5 pg 11: "and data stratification": what kind of stratification(s)? Please elaborate.

6. pg 15: "specimen types involved in an infection": how was it assured that the isolate/specimen came from an infection rather than from colonization?

7. pg 18 last para. "in JANIS, the total number..." total number of what? (isolates, patients, specimens, or what?)

8. pg 18 last sentence: "decreased ... up to xx%" suggest to write "decreased resistance rates from ... to.. "

9. pg 21 2nd para last sentence: "specimen-type specific infection rates" again how was it determined that samples originated from infections rather colonizations?

10.pg 23 para 2 first sentence: hard to understand what is meant.

In addition, the program code for the Java as well as for the Perl program should be made available.

Reviewer #2: The manuscript provides important information about 2 large surveillance programs. De-duplication is a very important issue, and the study provide a broad analysis on how it can affect the results of these large surveillance programs. The data is interesting and deserves publication, but the manuscript is too long, especially discussion and introduction. These 2 sections can be extensively reduced.

6. PLOS authors have the option to publish the peer review history of their article (what does this mean?). If published, this will include your full peer review and any attached files.

Reviewer #1: No

Reviewer #2: No

---

## [Author Response · Author response to Decision Letter 0]

5 Jun 2020

Point-by-point response to the editor’s and the reviewers’ comments: 

Editor’s comments:

The reviewers were positive about the manuscript. There are some changes to be done to increase clarity. The manuscript should also be reduced in the introduction and discussion. That will increase its clearness and bring a clear message.

Response: Thank you very much for your positive evaluation and constructive comments. We revised this article as below.

Response to Reviewer’s Comments

Reviewer #1: De-duplication of datasets used for surveillance of antimicrobial resistance is important and it has been shown that this strategy influences the final report. There are several strategies for de-duplication, the manuscript compares two of them, the Japanese JANIS and the WHO GLASS system. Depending on definition of the denominator for some antibiotics significant differences were found between the two methods. The methods and datasets appear comprehensive, presentation of results in general is adequate.

I only have few comments:

1. pg. 3 "per multi-resistance phenotype". In the whole paper there is no definition of multi-resistence phenotype. It only multi-resistant strains are included, the result is severly skewed. Explain or define "multi-resistance".

Response: Thank you very much for your comment. We removed the vague word “multi-resistance” throughout the text, and modified the sentences as follows: “removing repeated isolates of the same bacterial species isolated from a patient within 30 days, regardless of specimen type, but distinguishing isolates with change of antimicrobial resistance phenotype” (Abstract) and “Basically, JANIS uses a method that removes repeated isolates of the same species isolated from a patient within 30 days, regardless of specimen type, but still distinguishing isolates that showed change of antimicrobial resistance phenotype (e.g., from susceptible to resistant for any antibiotic, as detailed in Materials and Methods).” (Introduction)

2. pg. 8 line 2: "using"

Response: We revised the word accordingly (page7 line18).

3. pg 10 "considering antimicrobial resistance": in which form is resistance considered? This remains unclear in the whole manuscript.

Response: As explained above (No. 1), we modified it as “still distinguishing isolates that showed change of antimicrobial resistance phenotype (e.g., from susceptible to resistant for any antibiotic, as detailed in Materials and Methods)” (page 8 line 15-17). The detail in Materials and Method is as follows: “A subsequent isolate is selected and counted if it shows phenotypic change from susceptible to resistant (or vice versa) or a 4-fold or more comparison of MIC value for a specific antibiotic when compared to a previous isolate from the same patient within the 30-day period, as implemented in the in-house Java program. The server-side program of JANIS also selects the 1st isolate with intermediate susceptibility to a specific antibiotic isolated from the same patient within the 30-days period, although the intermediate susceptibility is not accounted for in this study because it focuses on and compares percentage of resistance.” (page12 line4-11).

4. pg 10 last line, pg 11 first line: usage of ~, -, <=, >= is not consistent in the whole manuscript, please change

Response: We revised the word accordingly as (<15, 15-64, 64<) (page10 line9) and (<1, 1-4, 5-14, 15-24, 25-34, 35-44, 45-54, 55-64, 65-74, 75-84, 84<) (page10 line10-11).

5 pg 11: "and data stratification": what kind of stratification(s)? Please elaborate.

Response: We elaborate it accordingly as “in each specimen type per age group, gender, and infection origin stratification” (page11 line6-7).

6. pg 15: "specimen types involved in an infection": how was it assured that the isolate/specimen came from an infection rather than from colonization?

Response: Thank you very much for this important comment. As suggested, there is no way to distinguish infection from colonization here, so we removed “involved in an infection” and modified the sentence as “older patients tend to have multiple specimen types such as blood, urine, and sputum” (page15 line7-8).

7. pg 18 last para. "in JANIS, the total number..." total number of what? (isolates, patients, specimens, or what?)

Response: We modified it as “total number of isolates subject to AST” accordingly (page18 line6).

8. pg 18 last sentence: "decreased ... up to xx%" suggest to write "decreased resistance rates from ... to.. "

Response: We fixed it as “decreased the resistance rates for blood isolates from 16.4% to 13.6% and …” accordingly (page18 line16).

9. pg 21 2nd para last sentence: "specimen-type specific infection rates" again how was it determined that samples originated from infections rather colonizations?

Response: We completely agree with your point as explained above (No. 6), and modified it as “infection or colonization” (page21 line10).

10.pg 23 para 2 first sentence: hard to understand what is meant.

Response: We removed the paragraph according to the comment of Reviewer 2 and Editor. What we meant was “Regarding the de-duplication method from JANIS, the 30 day rule is used to count the patients after removing repeated isolates, and also count the isolates by selecting those with different antimicrobial resistance phenotype even within 30 days.”.

In addition, the program code for the Java as well as for the Perl program should be made available.

Response: We have made the programs at https://github.com/bioprojects/GLASS-JANIS-comparison as now written in Materials and Methods (page10 line12-13).

Reviewer #2: The manuscript provides important information about 2 large surveillance programs. De-duplication is a very important issue, and the study provide a broad analysis on how it can affect the results of these large surveillance programs. The data is interesting and deserves publication, but the manuscript is too long, especially discussion and introduction. These 2 sections can be extensively reduced.

Response: Thank you very much for your comment. We shortened the introduction and the discussion accordingly. Specifically, we removed the following sentence.

Introduction-1:

For P. aeruginosa, Acinetobacter baumanii and other species commonly found in healthcare-associated infections, there are frequently many repeated isolates in multiple specimen types over an extended period time, which makes the impact of the different methods larger.

Introduction-2:

(isolates collected on hospital day 3 or later are considered “hospital origin” while other hospital isolates and all outpatient isolates are considered “community origin”): “For each surveillance period (e.g. 12 months), only one result should be reported for each patient per surveyed specimen type and surveyed pathogen” (Guide to preparing aggregated antimicrobial resistance data files in GLASS)

Introduction-3:

As mentioned above, the data submitted to GLASS were tabulated using a method that has been used in JANIS because the JANIS team had not yet developed a tabulation program implementing stratified de-duplication.

Discussion-1:

Regarding the de-duplication method from JANIS, the 30 day rule is used to count the patients after removing repeated isolates, and also count the isolates by selecting those with different antimicrobial resistance phenotype even within 30 days. When the latter is used as a numerator, the percentage of resistance increases slightly. This is because of the count of subsequent isolates that show change from susceptible to resistant (or vice versa) or 4-fold or more comparison of MIC values for an antibiotic compared to the previous isolate from the same patient even within a 30 day period However, in the data from Japan in 2017 used in the WHO GLASS report, the count did not account for the change of antimicrobial resistance phenotype and thus the percentage of resistance was closer to the value calculated using the first-isolate rule from GLASS.

Discussion-2 (moved to Supplementary Information):

This is now downloadable from the home page of the antimicrobial resistance research center at the National Institute of Infectious Diseases (https://www.niid.go.jp/niid/ja/from-lab/2415-amrc/9213-janis-glass-excel-en.html). Users can interact with these data by simply clicking their mouse. This tool (its graphical user interface is shown in S1 Figure) can create antibiograms stratified by the bed size of hospitals (under 200, over 200 and under 500, over 500 beds), inpatient or outpatient, gender, specimen type, and age group as defined by GLASS. Based on this, users can create antibiograms after excluding UNKNOWN_NO_AST. As another variable regarding missing data, GLASS defines UNKNOWN_NO_BREAKPOINTS representing the number of isolates where AST was performed but no interpretation of the results is available for a specific antibiotic. In the Excel tool, we have renamed this as UNKNOWN_NO_SIR to imply that S-I-R interpretation was not possible. It can be included or excluded from the denominator for calculating resistance rates in the tool.

---

## [Editor Report · Decision Letter 1]

15 Jun 2020

Comparison of de-duplication methods used by WHO Global Antimicrobial Resistance Surveillance System (GLASS) and Japan Nosocomial Infections Surveillance (JANIS) in the surveillance of antimicrobial resistance

PONE-D-20-00688R1

Dear Dr. Kajihara,

We’re pleased to inform you that your manuscript has been judged scientifically suitable for publication and will be formally accepted for publication once it meets all outstanding technical requirements.

Kind regards,

Patrick Butaye, DVM, PhD

Academic Editor

PLOS ONE
---

## [Editor Report · Acceptance letter]

18 Jun 2020

PONE-D-20-00688R1 

Comparison of de-duplication methods used by WHO Global Antimicrobial Resistance Surveillance System (GLASS) and Japan Nosocomial Infections Surveillance (JANIS) in the surveillance of antimicrobial resistance 

Dear Dr. Kajihara:

I'm pleased to inform you that your manuscript has been deemed suitable for publication in PLOS ONE. Congratulations! Your manuscript is now with our production department. 

Kind regards, 

on behalf of

Professor Patrick Butaye 

Academic Editor

PLOS ONE